# Apricot Rootstocks with Potential in Hungary

Edina Mendelné Pászti [1], Geza Bujdoso [2,*], Sezai Ercisli [3,4], Karoly Hrotkó [5] and Ákos Mendel [1]

[1] Research Centre for Fruit Growing, Institute for Horticultural Sciences, Hungarian University of Agriculture and Life Sciences, 2700 Cegled, Hungary

[2] Research Centre for Fruit Growing, Institute for Horticultural Sciences, Hungarian University of Agriculture and Life Sciences, 1223 Budapest, Hungary

[3] Department of Horticulture, Agricultural Faculty, Ataturk University, Erzurum 25240, Türkiye

[4] HGF Agro, Ata Teknokent, Erzurum 25240, Türkiye

[5] Institute of Landscape Architecture, Urban Planning and Garden Art, Hungarian University of Agriculture and Life Sciences, 1118 Budapest, Hungary

\* Correspondence: bujdoso.geza@uni-mate.hu

**Abstract:** In the last five decades, the use of rootstocks and scions has changed, along with their systems of cultivation. Associated with climate change, fruit trees face new ecological and phytopathological challenges. Rootstocks affect the generative and vegetative performance of a scion, such as productivity, span of nonbearing period, growth vigor, shelf-life and quality of fruits. According to a recent study, they also affect the frost tolerance of floral buds. Several traits of rootstocks facilitate the growth of a grafted tree under different climatic and soil conditions. Due to the high risks of cultivation, it is extremely important to determine which rootstocks are suitable for successful apricot production. Origin, effects on vegetative and generative traits, tolerance, resistance and adaptability of rootstocks are summarized in this review to select suitable rootstock for apricot cultivars under Hungarian conditions.

**Keywords:** compatibility; grafted tree; *Prunus armeniaca*; sustainability; usage; traits

## 1. Origin of Apricot

Most apricot cultivars belong to the *Prunus armeniaca* L. species (*Rosaceae* family, *Prunoidae* subfamily, *Prunus* L. genus), but other related species are also in cultivation in Asia. According to Vavilov [1,2] apricots originated in the highlands of northern China. The mountains of Tien Shan and Dzungaria are considered as secondary gene centers, as they are rich in wild relatives of apricot [3]. The Silk Road played an essential role in spreading it to other continents, which commenced from Xi'an (formerly Chang'an), and reached Byzantine, Venice, then Lyon [4]. In addition to fresh consumption, dried apricots are often sold, but several cultivars are also grown for their kernels in China [5].

## 2. Apricot Cultivation Worldwide and in Hungary

Apricot, after cherries and peach, are the third most economically significant stone fruit species in global production. Apricot is cultivated on 560 thousand hectares worldwide yielding up to four million metric tons of fruit every year. Mostly grown in regions with a Mediterranean climate, the fruits contain several substances that are important to human health [6]. Turkey, Iran, Uzbekistan, Italy, and Pakistan account for 54% of global apricot production [7]. More than 50% of production comes from Asia, followed by Europe (27%) and Africa (14%) [8]. Turkey produces the most apricots worldwide (695.000 tons in 2009). Apricots are generally sold as dried fruits, with only 2% sold on the fresh market. The most significant apricot producers after Turkey are Italy (233.000 t), France (190.000 t), Spain (97.000 t) and Greece (77.000 t) [7].

In Hungary, on average, 22.000 tons of apricots are grown on 5.000 hectares. The cultivation area is increasing by 100 to 200 hectares every year, resulting in an increase

in production. Consistent quantity and good quality of the product are indispensable for the long-term retention of fresh markets. It has been estimated that Hungarian producers could reliably sell 40.000 tons of fruit each year, but this has been revised to 50.000 tons. About 60% of the crop is sold as fresh fruit, and 10% is exported. The main export market is Austria, followed by Germany and Italy. The advantage Hungarian apricots have in the export market is their availability after Spanish, Greek and Italian production, when the European market is not saturated. Sales opportunities widened after joining the European Union; nevertheless, we can only be successful in these markets by maintaining strict quality requirements [7].

## 3. Propagation of Apricot and the Role of Rootstocks

Apricots can be propagated by seed but will not be true to type. Rootstock is only needed for budding or grafting to maintain and propagate the outstandingly worthy genotypes supervened during domestication. Budding and grafting are the most common vegetative propagation methods, while rooting of apricot cultivars (by cutting, layering or micropropagation) is difficult and inefficient. The importance of rootstocks is considerably due to spread of grafted trees [9,10]. Evaluation of apricot rootstocks is mainly based on the compatibility of the graft and the expansion of adaptability to soil conditions of the orchard, followed by other cultivating aspects such as vigor and resistance, etc.

Several traits of rootstocks (tolerance against pests, diseases, high lime content of soil) enable cultivation in locations that are not optimal for the needs of modern apricot cultivars. Rootstocks affect growth, vigor and phenology of the scion; quantity and quality of fruit; and tolerance to soil biotic and abiotic factors [11–15]. In addition, Nyujtó and Kovács [16] noted that some rootstock and scion combinations show higher resistance to bacterial canker. Some rootstocks affect tree performance via the development and extent of the root system, determining water and nutrient uptake and adaptability to ecological conditions (e.g., tolerance to frost, drought, calcareous soils, soil pH, salt, waterlogging, soilborne pests and diseases). Alternatively, rootstocks can affect the generative and vegetative performance of the scion through yield, growth vigor, quality and shelf-life of fruits [17–19].

In the past 50 years, apricot orchard production systems and cultivars have changed [20–22]. A group of the appropriate rootstocks for apricot cultivars is limited by compatibility.

## 4. Rootstock Breeding

During the evaluation of their largescale rootstock experiment, Southwick and Weis [23] identified that some Myrobalan rootstocks cause greater mortality in apricot orchards than in other species. Such incompatibility may not be exhibited for many years [24–26] but can be seen soon after grafting [27]. Several studies have focused on apricot rootstock incompatibility in recent decades, but the results cannot be generalized [28–30]. Compatibility tests are needed for every rootstock and scion combination. In addition, rootstock–scion combinations must be determined for geographical suitability [31].

New apricot rootstocks are required to combine resistance or tolerance against nematodes, diseases, pests and edaphic conditions, while good performance in the nursery and exceptional rooting is also essential. To capitalize on the potential of rootstock–scion combinations, the interdependence of their vegetative and generative traits must be determined accurately.

The breeding of new clonal apricot rootstocks occurs in many locations; often based on interspecific hybrids of plums, peaches or other stone fruit species, they can have advance rootstock breeding [27,32]. In the Czech Republic, apricot scion and rootstock breeding programs were initiated in the 1970s, and rootstock breeding also started in Romania at that time. Lower growth habit and good adaptability were the aims of the apricot rootstock breeding program in Pitesti (Romania) [33]. At the University of Novi Sad in Serbia, *Prunus cerasifera*, *P. spinosa* and *P. domestica* species were involved in the rootstock

breeding program [34]. In Turkey, a selection of interspecific hybrids of *P. armeniaca* × *P. salicina* cv. 'Black Amber' were assessed for incompatibility [35]. A large number of interspecific hybrids were produced in Ukraine from *P. armeniaca* and wild Middle-Asian *Prunus* species crosses. Rootstocks of 'Alab 1' (*P. cerasifera* × *P. armeniaca*), 'Druzhba' (*P. pumila* × *P. armeniaca*) and 'VVA1' (from Krymsk series) came from this research, and several apricots and plumcot scion cultivars were also released [36].

In Hungary, where seedling rootstocks are still dominant in nursery production, a seedling rootstock selection program was operated at the Cegléd Research Station [37]. Subsequently, a clonal rootstock evaluation trial was established in Cegléd over the last decade [38].

## 5. Grafting Compatibility of Apricots

Although commonly used rootstocks are compatible with most of the apricot cultivars, this trait is not always evident [39]. Grafting compatibility of the apricots with apricot seedlings (*P. armeniaca* L.) is excellent; no exceptions can be found in the literature. Only chlorotic leaf spot virus (CLSV) can cause compatibility problems on apricot seedling rootstocks [40]. First signs of this phenomenon are the poor graft union formation in the nursery and the death of shoots. Symptoms are more significant when only one of the grafting partners is infected, mostly when the virus-free rootstocks are budded with infected scions. *Phytoplasma* contamination of a component results in similar symptoms.

Apricot cultivars usually have good-to-moderate compatibility with plum species (*P. cerasifera* Ehrh., *P. salicina* Lindl., *P.* × *mariana*, *P. insititia* Jusl., *P. domestica* L.). The incidence of incompatibility with plum and Myrobalan species occurs more often when grafting, rather than during the budding of an apricot scion. In these cases, parenchymal cells are stuck between the tracheas and the stereomes [24,41,42], and mechanical forces (such as strong winds) can separate the smooth surfaces between the scion and the rootstock [43]. This sign of incompatibility is often partly on the surface of the graft union or occurs in the later life of the graft (Figure 1).

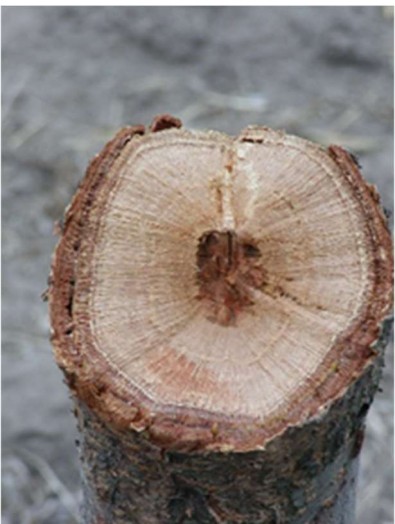

**Figure 1.** Effect of incompatibility of apricot cultivar and plum trunk (Photo: Hrotkó).

Reig et al. [42] observed that incompatibility most often exhibits within the first four years of growth. Reduced thickness in the scion compared to the rootstock (by 20% or more) indicates incompatibility, signaling a discontinuity in the phloem or the xylem. Irisarri et al. [44] isolated genes encoding phenylalanine ammonia-lyase (PAL), involved in trachea differentiation and in the metabolism of phenol derivatives. Expression of PAL genes was different in compatible and incompatible combinations. Excessive water and nutrient supply in young trees can increase the incidence of incompatibility, thought to be associated

with rapid growth and trunk thickening, whereas during the early years of an orchard, summer pruning and the moderation of water and nutrient supplies can lower risk. In France, apricots grafted on plum rootstocks are often budded at 40 cm height, as incompatibility is less common with long rootstock shanks.

In southern France, apricot is usually budded on peach (*Prunus persica* L.) and almond (*P. amygdalus* L.) seedlings, or on their hybrids. Sand cherry species (*P. pumila* L. and *P. besseyi* Bailey) and their interspecific hybrids are moderately suitable as rootstock for apricots, but downy cherry (*P. tomentosa* Thunb.) seedlings show incompatibility symptoms [18]. Complex interspecific hybrids, selected by Eremin [36] in Crimea, show high incompatibility rates or poor fruit quality [45–49]. These rootstocks often show difficulty in the translocation of nutrient elements, with root starvation leading to slow tree death. Apricot cultivars are usually compatible with peach seedling rootstocks, but in reverse, translocational disorders often occur [24,27,50].

## 6. Environmental Adaptation of Rootstocks

Apricot orchards are rarely established in locations with optimal environmental conditions; therefore, adaptation is a crucial factor in selecting and breeding of apricot rootstocks.

### 6.1. Soil Requirements

In areas of lower altitudes, with colder, wet and heavy soil, apricot seedlings and plum rootstocks are more common in apricot production. Suitability of the environment is the main factor in the location of apricot orchards; therefore, the adaptability of the rootstock to soil conditions is the main priority. Heavy, moist and cold soils require rootstocks with good tolerance against waterlogging. Stagnant waters are lethal to apricot seedlings [50]; clonal Myrobalan and 'Brompton' rootstocks are moderately tolerant, while bullace, Damascene and Marianna plum have full tolerance against these soil conditions.

Apricot seedlings tolerate high lime content soils well, originating from northwest China, where these soils are common. Apricots grafted on peach seedling rootstocks growing on high lime soils exhibit symptoms of severe leaf chlorosis. Lime tolerance of Myrobalan species is even lower than that of apricots, while Marianna GF 8-1 is better in this aspect. The Spanish bullace rootstock cultivar 'Pollizio' has the best adaptation to calcareous soils among the plum species [27]. Despite these results, in trials in Szigetcsép (Hungary) on calcareous soil, apricots grafted on C. 162 Myrobalan seedlings have more healthy foliage than those on C. 1650 apricot seedlings [51].

Important differences have been observed in the presence of natural mycorrhiza colonization between apricot rootstocks from different species. A higher rate of colonization aids tolerance to abiotic stresses through amelioration of water and nutrient uptake. Apricot and Myrobalan seedlings are widely used in Hungary, and these rootstocks have had the best receptivity so far. The less commonly used Montclar, GF 677 and Rootpac R rootstocks had moderate affinity [52].

### 6.2. Cold Tolerance and Winter Hardiness

In the northern part of Europe, efficient apricot production is more dependent on the winter hardiness of the rootstock–scion combination. Apricot seedlings have a wide range of tolerance to freezing winter conditions, associated with origin of the rootstock. In northwest China, apricot is produced in regions with long winter periods, with minimum temperatures below −20 °C for several days [53]. Layne and Harrison [54] stated that the 'Haggith' cultivar, in addition to providing a uniform seedling progeny, has solid hardiness in Canada, and based on the experiments of Kappel [55], using this rootstock provided the best apricot yields. Select *P. armeniaca* rootstocks possess winter hardiness of both roots and trunk. However, where scions are damaged by freezing winter temperatures, budding at height is often employed. A few plum scions are suitable for this purpose, e.g., Brompton is used in Germany, Buduruz is common in Romania for trunk bearing. In Hungary, Fehér besztercei has produced positive results with this method.

*6.3. Susceptibility to Pests and Diseases*

The most adverse pests of rootstocks are nematodes, with *Meloidogyne* spp. being the most prevalent in the Mediterranean apricot-producing region. Apricot seedlings tolerate these species well, whereas the most common rootstocks in southern Europe (peach and almond species) are susceptible. Myrobalan B, Marianna GF 8-1, and GF 31 plum rootstocks are somewhat tolerant to *Meloidogyne* spp. In northern regions, *Pratilenchus* species are more widespread; apricot seedlings GF 31 Myrobalan and GF 2038 hybrids show tolerance; other rootstocks were proved susceptible. *Xiphinema* spp. plays a unique role in the spread of viruses; thus, nurseries and virus-free plantations should only be established at nematode-free locations.

Among the diseases of apricots, a select number of rootstocks are sensitive to *Verticillium*, with infection causing dieback. GF 31 is particularly susceptible, while Marianna GF 8-1 plum and GF 1380 greengage have moderate tolerance. Bacterial canker (*Pseudomonas syringae* and *Pseudomonas mors-prunorum*) causes damage to the bark of trunks and main branches during mild winters and spring. Plum species are the most sensitive to *Pseudomonas* spp.; apricot and peach seedlings possess some tolerance. This disease rarely appears in trees that have been budded at 40–60 cm height. Root gall (*Agrobacterium tumefaciens*) is quite prevalent in apricot orchards and in nurseries. Apricot, peach and almond seedlings are susceptible to *Agrobacterium*. Rubira peach, Marianna GF 8-1, Myrobalan GF 31 and GF 1380 greengage are known for their moderate tolerance to this bacterium [27].

## 7. Rootstock Effects on Vigor and Productivity

Ten percent of Hungarian apricot orchards are intensively planted (distance between and within rows is less than 5 m and 3 m, respectively), and 60% of them are irrigated. Trees in these orchards are typically trained to open vase (45%) systems, or have a natural round shape, but intensive compact vase (10%) and spindle (5%) training are also practiced [56].

Increasing planting density from medium (600 to 750 trees/ha) to high density (1.000 to 1.250 trees/ha) results in substantial changes to orchard management for orchardists [57]. Upright axes training systems are more productive; therefore, where possible, canopies should be created with at least one or more axes in commercial orchards [58–62]. Using these canopy forms, a planting density of 1.100 to 1.600 trees/ha can be achieved.

Wild apricot and Myrobalan are widely used as rootstocks in more countries [6,63]. In orchards with these rootstocks, pruning is important in managing the size of the canopy. Pruning during winter and summer needs to moderate the canopy growth of the grafted tree to encourage more flower buds and enhance light exposure of fruit to ensure high quality [64]. In Hungary, experimental orchards with spindle canopy were established at the predecessor of the Hungarian University of Agriculture and Life Science in the 1990s (Figure 2).

According to experience in France, the strongest vigor was observed on cultivars grafted on clonal Myrobalan rootstock and their hybrids (GF 31, GF 8-1), as well as on apricot seedlings. Sequentially lower vigor was expressed on greengage, medium–semi dwarf plums (e.g., Saint Julien). Monney et al. [65] published results about reaching 35–45% dwarfing effect on Citation, Rubira and Jaspi, but GF 655-2, W61, Pixy and Torinel decreased the vigor of the grafted trees by only 15 to 25% compared to Myrobalan in Switzerland. True dwarfed trees can be created on *Prunus besseyi* (GF 2037 and GF 2038). There is a question among growers as to how important it is to create dwarf trees, as rootstocks with high vigor can be hand-harvested, while industrial orchards employing mechanical harvesting also benefit from strong-vigor rootstocks [66].

With regard to yield, there are large differences among seedling rootstocks. In general, cultivars grafted on apricot seedlings produce many flowers and fruits; however, these trees tend to bear late, and fruit size can be small. Fruit size harvested from trees grafted on Myrobalan and plum rootstocks is better compared to the fruit size harvested from trees grafted on apricot seedling rootstocks. Frost damage during the flowering period can result in less reduction of yield for trees grafted on seedling rootstocks [15].

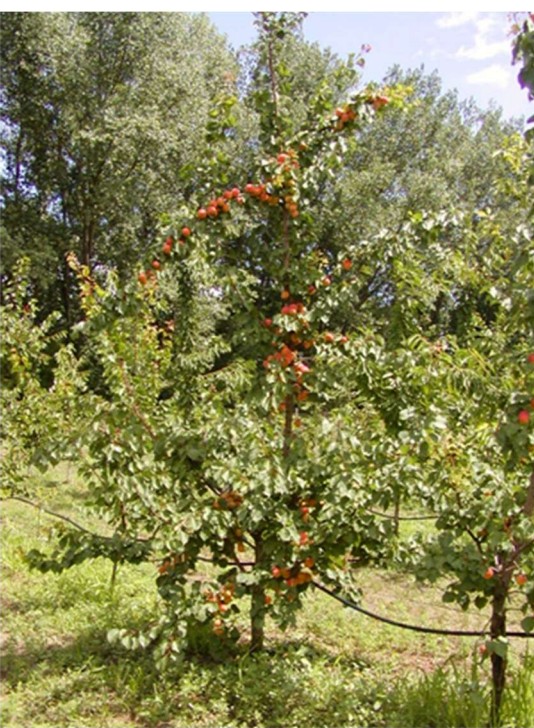

**Figure 2.** An example of a 6-year-old 'Bergeron' apricot grafted on Marianna GF 8-1, trained to a spindle canopy (Photo: Hrotkó).

Trees grafted on plum rootstocks start sap flow earlier compared to trees grafted on apricot seedling rootstocks, so they are more sensitive to spring frost events. Winter freeze damage can occur in one-year-old shoots of scions grafted on high vigor rootstocks (Rootpac R, Montclar) under Hungarian climatic conditions, where shoot lignification is delayed [67]. In the production regions of Turkey, the fruit sugar content of trees grafted to dwarfing rootstocks (e.g., Pixy) and apricot seedlings is high, and is more suited to dried fruit production, whereas scions on Myrobalan have better fruit size, more suited to production for the fresh market [68].

## 8. Rootstocks in Hungary and around the World

In Hungary, 73% of apricot trees are on Myrobalan (seedling and clonal combined), another 10% are on apricot rootstocks. The ratio of the vegetative plum rootstocks is 17% [56]. In Hungary and its neighboring countries, old orchards are typically on apricot seedlings, but in recent decades, clonal Myrobalan rootstocks have predominated [69,70]. Similar to Hungary, there were rootstock breeding programs in the Czech Republic (M-VA) and Romania (Constanza 14, Constanza 16) [71,72]. There are many countries around the world where Myrobalan and its vegetatively propagated hybrids (Myrobalan B, Myrobalan 29C) are used as rootstocks. The Myrobalan 29C used in larger quantities during the past decades provides this rootstock an early bearing period and good adaptation, especially on soils with high lime content [73]. Some rootstocks increase the yield of grafted trees, such as 'Marianna GF 8-1', 'Greengage CD-4', and 'Damas1869' rootstocks [74]. 'Marianna GF 8-1' provides long life for the grafted cultivars. The 'Pollizo' plum rootstock (*Prunus insititia* L.), derived from Spain, has good tolerance to anaerobic soil conditions, which is typical for Mediterranean regions [50]. Studies have confirmed that 'GF 305' (a peach seedling) and Real Fino (apricot seedling) rootstocks are resistant to apple chlorotic leaf spot virus (ACLSV) [75]. 'Wavit', derived from the Wagenheim plum seedling population (*Prunus domestica* L.), is well known for uniform orchards and decreased vigor; the use of this rootstock for apricot is increasing in areas with cold and humid conditions. The rootstock 'Penta'

was bred in Italy; this is a cross between 'Tetra' (*Prunus domestica*) and 'Adara' (*P. cerasifera*). 'Plumina' is a French interspecific hybrid from the cross *Prunus besseyi* × *P. cerasifera* [76].

The vigorous 'Monclar' (*Prunus persica* L.) rootstock supports the early turning to bear of grafted apricot and peach cultivars [77]. 'Rootpac R' has characteristics similar to those of 'Monclar', with good adaptability to suboptimal soil and climatic conditions and strong vigor. It is derived from a *Prunus cerasifera myrobalana* L. × *Prunus dulcis* Mill. cross, and it is mainly used for replanting [78].

### 8.1. Wild Apricot (Prunus armeniaca L.)

This rootstock is propagated by seeds only. There are currently no known apricot clones that can be propagated at an economic scale using either cuttings or in vitro methods. Apricot seedlings have good compatibility with apricot cultivars producing high- or very high-vigor trees. Rootstocks named wild apricot can include seedlings of cultivars with small fruit size (e.g., 'Korai piros'), local cultivars (e.g., in North Africa 'Balady' and 'Mech-Mech'), and also the mostly wild, so-called sea apricot rootstocks. In almost all central European countries, a selection of sea apricot mother trees for seedling production exist, selected because their seed to fruit ratio is good, their germination ratio is high, and their seedling plants are very uniform in the nursery. In France, a local cultivar called 'Manicot' ('GF 1236') has strong vigor and produces a homogenous population of rootstocks [79,80]. The self-fertile seed-bearing cultivar of 'Haggith' was selected in Ontario, Canada, with vigor stronger than 'Manicot'. 'Haggith' has good winter hardiness and is not susceptible to diseases that attack apricots. Seed-bearing mother trees producing seeds for rootstocks such as 'M-VA-1', 'M-VA-2', 'M-VA-3', 'M-VA-4', selected in Czech Republic, have seedlings of similar quality to 'Haggith' [72,81]. From Romania, Indreias and Trandrafirescu [82] selected some seed-bearing apricot genotypes. In Hungary, the seed-bearing apricot rootstock breeding program derived from apricot seedlings (such as 'C. 1301', 'C.1650', 'C. 1652') is run by Prof. Nyujtó at the Research Station of Cegléd [83–85]. Cultivars grafted on these rootstocks bear well; however, the yield efficiency index on them is low compared to other rootstocks, partly because of their strong vigor. Trees grafted on apricot seedling rootstocks prefer light soils with neutral or mildly alkaline pH and good water capacity [86]. Apricot seedling rootstocks are sensitive to waterlogging, but they tolerate the high lime content of soils well [51]. Wild apricot rootstocks have excellent compatibility with all apricot cultivars within the species [48] but are very sensitive to *Vertcillium* and CLSV; therefore, it is recommended to bud or graft virus-free selections [79,80].

### 8.2. Plum Species as Rootstocks for Apricots

European plum species as rootstocks for apricots have become very popular among growers in recent decades, as they have a good grafting rate [51], although there are differences between the various species and groups. The taxonomical questions of the European plum species are very difficult; therefore, it is necessary to create some groups to orientate among them. The most important plum rootstocks and hybrids can be put into four groups:

1. Myrobalan plum and its hybrids;
2. Bullace and Damascena plum;
3. Local European plum and the local selected plum cultivars;
4. Other species and interspecific hybrids [86].

### 8.3. Myrobalan (Prunus cerasifera Ehrh. var. cerasifera Schneid. cv. myrobalana)

Myrobalan is native to Europe and Asia, with a widespread range, and it is planted as both a rootstock and ornamental tree. Hungarian nurseries graft 70% of apricot cultivars on Myrobalan [56]. Myrobalan is a very diverse subspecies; it survives high soil water levels and waterlogging but does not like dry, stony soils. Apricot cultivars grafted on it are susceptible to late spring frost damage and to *Verticillium*. Incompatibility depends on their origin; among the seedling Myrobalan rootstocks derived from Cegléd, only 'C. 174' exhibits

incompatibility and only in the case of apricot. Scions grafted on Myrobalan produce strong vigor, grow faster, turn to bear early and their yield is higher compared to trees grafted on wild apricot seedlings. Among the Myrobalan rootstocks derived from the breeding program running at the Research Station of Cegléd, 'C. 162' and 'C. 359' are recommended as rootstocks for apricot [69,83,86]. The Myrobalan 'C. 162' rootstock is not suitable for select important cultivars ('Ceglédi kedves' and 'Ceglédi bíbor') [51,83,84,87–90].

### 8.4. Vegetatively Propagated Myrobalan Species

'Myrobalan B' is one the oldest clonal Myrobalan rootstocks, and it was selected in East Malling. The cultivars grafted on it produce very strong vigor and begin to bear late. This rootstock grows well in all types of soil. In orchards, it produces few suckers. Its compatibility with Hungarian-bred cultivars is unknown at present; according to Magyar [51], 'Bergeron' and 'Ceglédi óriás' grafted on it had a low survival rate in the nursery. In long-term trials under Hungarian climatic conditions, this rootstock produces good results. It has tolerance to *Meloidogyne* sp. when used as interstock and is known to be resistant to *Pseudomonas* sp. [27,89,91].

In recent years, the use of 'Myrobalan 29C' is increasing in Hungary. This rootstock was selected by Gregory Brothers nursery in Brentwood (CA, USA). In the USA and Italy, this rootstock can be grafted with both peach and almond. Propagation is achieved using softwood cuttings or micropropagation; it has strong vigor, but its root system cannot fix the whole tree in the soil, requiring support. 'Myrobalan 29C' adapts well to different soil conditions. Its rate of sucker production is moderate. Compatibility with different apricot cultivars is very good. According to Grassely and Day, this rootstock is a hybrid of Marianna plum rootstock [27,91,92].

### 8.5. Bullace, or St. Julien Plum (Prunus insititia Jusl.)

These penta- or hexaploid species, which have round-shaped fruits, are native to Hungary. They have strong vigor but are considered a medium-vigor rootstock. It is important to mention that there are dwarf rootstocks within this group. According to the newest results, bullace is a natural cross of sloe (*Prunus spinosa* L.) and cherry plum (*Prunus cerasifera* Ehrh.) that may have occurred in the overlapping area of both species in Europe [93,94].

The oldest St Julien clone is 'St. Julien A', which was selected in East Malling. Initially, this rootstock was selected for plum, but it has good compatibility with apricot scions, inducing medium-vigor trees. As a young standalone tree, it produces upright shoots with few laterals. Cultivars grafted on this rootstock produce strong, vigorous trees; however, the canopy volume of the grafted trees is only 75% of 'Myrobalan B' [27,91,95,96]. The 'INRA Saint Julien GF 655/2' clone is used as an apricot rootstock outside of France. This rootstock can be propagated using soft- and hardwood cuttings, and it is a good rootstock for plum, peach as well as apricot cultivars. In the orchard, this rootstock does not produce suckers. It is not sensitive to gall formation or bacterial canker and tolerates limy and dry soils. Semidwarf or medium vigor is induced in grafted scions. This rootstock tolerates replanting and limy soil conditions well, and has good winter hardiness under Hungarian conditions. In the nursery, 'INRA Saint Julien GF 655/2' produced a good compatibility rate; 'Bergeron', 'Magyar kajszi C 235' and 'Ceglédi óriás' grafted on this rootstock achieved medium vigor [27,51,80,80,91,97].

One of the most promising bullace rootstocks is 'Adesoto 101'$^{\circledR}$, derived from Aula Dei Research Station (Zaragosa, Spain). It has excellent compatibility with peach, apricot, almond and plum cultivars [40,98]. It is not susceptive to root gall or iron chlorosis; it tolerates high pH and 10 to 11% active lime content in soil well [99,100]; it is also tolerant to drought. Vigor of the cultivars grafted on this rootstock is approximately 80% of scions grafted on 'St Julien A' and peach seedling rootstocks [101]. 'Adesoto 101'$^{\circledR}$ is resistant to *Meloidogyne* species, but its vigor depends on *Pratilenchus vulnus* infection. 'Adesoto 101'$^{\circledR}$ is not very susceptible to *Pratilenchus vulnus* compared to 'St. Julien GF 655/2' and 'Citation'

rootstocks, which are very susceptible [102]. 'Adesoto 101'® showed incompatibility among Italian conditions in heavy soils [76]; the 'Adara' rootstock was weaker in heavy soil compared to 'Adesoto 101'®.

*8.6. European Plum (Prunus domestica L.)*

Hungarian fruit growers often graft or bud plum and apricot cultivars on different local plum varieties, seedlings and their suckers. Some of these rootstock genotypes, derived from the local plum cultivars, are used in foreign nurseries. In Hungary, 'Fehér besztercei' and 'Kisnánai lószemű' were released as state-approved cultivars, which had been selected for apricot from local plum varieties. Other plum seedlings such as 'Vörös szilva' and 'Bódi szilva' are older selections. There is a lack in Hungarian fruit research, as the previously mentioned two rootstocks are not introduced in production.

'Fehér besztercei', was bred by Dr. Pál Nagy in the 1960s from a selection of the local population at the Research Institute for Fruit Growing (Budapest, Hungary) [103]. It is a state-approved cultivar that grafts well to 'Magyar kajszi', but this rootstock has good compatibility with other cultivars. According to the results derived from trials set up at the late University of Horticulture and Food Industry (Budapest, Hungary), this rootstock is considered ideal for apricot. It can be propagated using hardwood cuttings, with cuttings collected in December producing the best rooting rate [104,105]. Young rootstock plants are susceptible to dry conditions, and in the first few years, young plants need careful nursing and regular irrigation. In the nursery, the young plants start to grow slowly, but despite this, the survival rate of the trees grafted on 'Fehér besztercei' is good, though their vigor is weaker compared to trees grafted on wild apricot and Myrobalan seedlings. Canopy volume of the trees grafted on 'Fehér besztercei' is 35 to 40% smaller than the trees grafted on seedling rootstocks, but their yield efficiency is 20% more. The rate of trees that died from apoplexy is 50% less compared to trees on wild apricot seedling rootstock [103]. The 'Kisnánai lószemű' plum was selected from a local population by Dr. Pál Nagy and his research group. This rootstock was used as a rootstock of the apricot cultivar 'Borsi féle kései rózsa' in the trials.

Two European plum rootstocks, 'Penta'® and 'Tetra'® [106], have compatibility with both peach and apricot cultivars. Both induce semidwarf–dwarf trees and are useful rootstocks around Ancona (Italy) [107]. However, some authors have identified insufficient quantity of mature trees after the seventh leaf [76].

In Hungary, the rootstock 'Wavit' appeared in some trials, and this rootstock needs more attention. 'Wavit' is an in vitro-propagated clone of the rootstock 'Wangenheim', which is an old German plum cultivar with medium fruit size, and its fruits ripen in late August. Its seeds germinate well and produce a uniform progeny. Nurseries know it as a cultivar suitable for creating good progeny. In Poland, plum scions, trained to spindle canopies, grafted on this rootstock, produce medium–semidwarf vigor [108–110]. However, the apricot cultivars on 'Wavit' have dwarf vigor, small fruit size and high tree mortality [111,112]. As an apricot rootstock, it is useful in northeast Austria in heavy soil, resulting in dwarf trees suitable for intensive orchards, but Wurm [49] observed small fruit size in cultivars grafted on 'Wavit'.

'Bromton', from the United Kingdom, and selected from local cultivars, is often used as interstock in western Europe, but it is also known as rootstock in Europe too. It has good compatibility with grafted scions and produce trees with medium-to-strong vigor. This rootstock prefers heavy soils, but it is susceptible to gall [27,80,91]. As a rootstock, 'Bromtom' results in a tree with a good trunk when used as interstock; often, on Myrobalan seedling rootstock, high trunk grafting is used. In France, the clone 'Reine-Claude GF 1380' was used as interstock for apricot. Today the 'Torinel'® (Avifel) gage type is increasing in use [48]; it is derived from a *Prunus domestica* P994 x 'Reine Claude de Bavay' Nº24 cross. 'Torinel' is recommended for specifically wet and heavy soils; it is not susceptible to gall formation. It has good compatibility with apricot cultivars, which bear fruit faster on 'Torinel', than on 'Reine Claude GF 1380' under the same conditions.

### 8.7. Interspecific Hybrids

Among Hungarian climatic conditions, 'INRA Marianna GF8-1' is a suitable rootstock for apricot. This rootstock is derived from a cross between Marianna plum and Myrobalan from the research station in Grande Ferrade (France). As rootstock for apricot, this stock produces a strong-vigor tree, and according to the French experience, this rootstock has good compatibility with 'Bergeron', but its compatibility with 'Canino' and 'Rouge de Roussillon' cultivars is not satisfactory. Apricot cultivars grafted on 'INRA Marianna GF 8-1' had strong vigor in the trial planted in Szigetcsép (central Hungary); 'Magyar kajszi C235', 'Bergeron' as well as 'Ceglédi óriás' were grafted onto this rootstock, which began cropping early. Under Hungarian climatic conditions, it has good winter hardiness and adapts well to a range of soil conditions. Its resistance to *Pseudomonas* is good, it is not particularly susceptible to *Phytophtora*, it is resistant to *Meloidogyne* and *Armillaria mellea*, and is tolerant to *Verticillium*. This rootstock is susceptible to silver leaf (*Condostereum purpureum*); therefore, all tools must be disinfected in the nursery. It is a strong, vigorous rootstock; its root system explores the soil widely but does not produce suckers [27,51,79,80,89–91].

'Ishtara' [®] (Ferciana) rootstock was bred from crosses of Myrobalan and peach (P. 322 X P 871/1) in Grande Ferrade (France). At first, it was used as a rootstock for plum, but it has good compatibility with Japanese plum, apricot, peach and almond cultivars too. Its vigor is weaker compared to Myrobalan and is considered a medium-vigor or semidwarf rootstock. Cultivars grafted on 'Ishtara' crop early and produce well, while it has a good tolerance to bacterial diseases [113,114].

'Jaspi [®] Fereley' (*Prunus salicina × spinosa*) is a strong vigorous interspecific hybrid that can be grafted to both apricot and peach. Cultivars grafted on this rootstock crop early and yield well. It is not susceptible to waterlogging. New-bred Spanish rootstocks such as 'Miragreen' (*P. cerasifera × P. davidiana*) and 'Mirared' (*P. cerasifera × Nemared*) showed good compatibility in 90% of propagated apricot cultivars [115].

In Romania, apricot cultivars grafted on 'Saint Julien A', 'Otesani 8', 'Scoldus' and 'Miroper' rootstocks provided trees with the longest lifespan [116] and strong vigor. They are crosses between Myrobalan and a peach hybrid. Peach (*Prunus persica* L.), almond (*Prunus amygdalus* L.) and their hybrids (*Amygdalopersica* and *Davidiopersica* hybrids) are not used as rootstocks for apricot in Hungary [67], although these rootstocks are used by French ('Montclar', a 'Rubira'), American and Israeli producers with great success. In Israel, almond rootstocks (Alnem series) are used in very dry soils with high lime content, and bitter almond was used as an apricot rootstock on fruit sites with similar conditions in Hungary during the 1930s. 'Nemaguard' is used in the USA for its tolerance to dry soils with high lime content and ability to tolerate nematodes (replanting), while 'GF 677' with an interstock is used in Italy.

Sand cherry (*Prunus besseyi* Bailey, *P. pumila* L.) is more closely related to plums than cherries, and can be used as rootstock for peach, apricot and plum. Some genotypes from the eastern sand cherry (*P. pumila* L.) and western sand cherry (*P. besseyi* Bailey) and their hybrids were selected as vegetatively propagated rootstocks with dwarfing vigor. 'Pumiselect'[®] from Giessen (Germany) is a good rootstock not just for plum but also for peach and apricot cultivars. There was a low tree survival rate on 'Pumiselect'[®] in Bulgaria [117]. In France, some dwarf genotypes ('Prumina'[®], *P. besseyi × P. cerasifera*) for peach and apricot were selected from hybrids of western sand cherry. These rootstocks, originating from a cross of *P. besseyi × P. cerasifera*, are under evaluation in trials across Italy. Initial results suggest these rootstocks have dwarf vigor, but more data are required to evaluate their production [76]. Recently, 'Krymsk 1'[®] (VVA-1), a dwarf-interspecific hybrid (*P. tomentosa × P. cerasifera*) from Crimea, has been adopted in USA. This rootstock results in improved yield and a smaller tree size than trees grafted to 'St. Julien A'. 'Krymsk 1'[®] is susceptible to high lime content; older age incompatibility can be observed with some cultivars [45–47]. Symptoms of incompatibility on cultivars grafted on 'Krymsk 86' have been noted under Spanish climate conditions [48].

### 9. Suckering

Suckering of apricot rootstocks have high correlation with plum pox virus infection, as new shoots attract aphids. In addition, the cost of orchard management increases, as the removal of suckers is labor intensive [23]. Knowles et al. [118] found that 'Marianna 9.52' produced an unmanageable number of suckers, while the propensity of 'Marianna 6.64', 'Marianna GF 8-1', 'Marianna 2624', 'Pixy', 'Ishtara' and 'St. Julien GF 655/2' to sucker was moderate.

*Prunus armeniaca*, *Prunus persica* and *Prunus dulcis* produce no suckers, whereas species of *Prunus sect. Prunus* (*P. cerasifera*, *P. domestica*, *P. insiticia*, *P. salicina*, *P. spinosa*) have high suckering potential [23,119,120]. *P. pumila* and *P. besseyi* also sucker [121,122]. It seems to be a hereditary dominance effect of suckering, as most of the hybrids of suckering species have this trait. Every plum, Myrobalan, bullace, greengage and blackthorn rootstock produces suckers.

### 10. Conclusions

Apricot is a significant stone fruit species cultivated worldwide. Most of the 560,000 hectares of orchard is located in warmer climates. In modern fruit growing, the importance of rootstocks has expanded considerably, as grafted trees are used to achieve production goals. The lack of a wide range of adaptability (which is typical of apricot) means that the rootstock is extremely important for plantation establishment. During the last 50 years, more than 100 rootstocks have been developed for apricots, each with different growth vigor and tolerance against biotic and abiotic factors, and they originate from several species. Virus-free, homogenous and uniform planting material is required for a good orchard, and this only can be ensured using micropropagated rootstocks. Interspecific rootstocks can provide added value. In practice, only 6–8 rootstocks are widely used by producers, even though new selections arguably offer better performance. This low number is due to a lack of practical experience. Largescale comparative experiments could help to select appropriate rootstocks. Available rootstocks could be mentioned for specific growing areas based on the generated data to help apricot producers.

Most of the commercial apricot plantations in Hungary are established on 'Myrobalan 29C'. Based on experiments and experience so far, vegetatively propagated Myrobalan rootstocks are well suited to Hungarian climatic and soil conditions, providing that tree water requirements can be met. However, in calcareous soils with poor water management, rootstock with strong root growth helps maintain the performance of the orchard. A greater adoption of rootstocks of interspecific hybrid origin with strong vigor can be expected in the future.

**Author Contributions:** Conceptualization, Á.M. and E.M.P.; methodology, K.H. and S.E.; formal analysis, S.E.; investigation, K.H.; writing—original draft preparation, Á.M. and E.M.P.; writing—review and editing, G.B., K.H. and Á.M.; supervision, G.B.; funding acquisition, G.B. All authors have read and agreed to the published version of the manuscript.

**Funding:** This research received no external funding.

**Data Availability Statement:** For the preparation of this review, no new, unpublished data were created.

**Conflicts of Interest:** The authors declare no conflict of interest.

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
