# Peer review of "Apricot Rootstocks with Potential in Hungary"

_horticulturae, doi:10.3390/horticulturae9060720_

Round 1
Reviewer 1 Report
The paper is very good. My suggestion are in attached pdf file.

Author Response
We have taken on board your comments and have rephrased the relevant parts based on your suggestions. The English of the manuscript is now more precisely fit in the academic language. Your review was very helpful and effective.Reviewer 2 Report
Comments:
The manuscript titled ‘Apricot rootstocks and their developmental opportunities’ by Pászti et al. discusses the opportunities and constraints of apricot rootstocks. However, the manuscript content appears to be poorly organized and not too sound academically. Authors are kindly suggested to address the comments offered below and modify the manuscript. (Note: the manuscript may require changes to its academic writing more than suggested here. The authors are requested to get help from someone professionally proficient in English from the relevant field of study.)
Major comments:
1. As mentioned above, the authors are kindly requested to use more academically sound and precise terminologies while explaining. Some of the ambiguous phrases are-
a. ‘hectic behavior of the climate’ (Page 1, line 15)
b. ‘spreading of engraftments’ (Page 2, line 65; Page 12, line 512-513)
c. ‘similarly poor obliteration’ (Page 4, line 126-127)
d. ‘Breaking down of an incompatible apricot…’ (Figure 2: Page 4, line 137)
e. ‘yielding’ (Page 6, line 223)
2. The manuscript appears particularly focused on the state of apricot grafting in Hungary. The authors are suggested to modify its title accordingly.
3. The abstract is expected to convey the gist of the manuscript in a concise manner instead of representing the background of the study. The authors are kindly suggested to modify/rewrite the abstract accordingly.
4. The contents under the headings 4, 5, and 7 seem very similar and/or overlapped. Authors are kindly suggested to either merge them or separate their contents so that their message would not appear redundant.
5. Inclusion of authors’ perspective on potentially best rootstock for Hungary (or to the part of it) would benefit the manuscript. The authors are kindly suggested to do so.
Minor comments:
1. The legend of Figure 1 does not seem sufficient. The authors are suggested to elaborate more on the intended message in reference to the signs, symbols, and lines in the figure.
2. Page 4, line 129-131: Sentence ambiguous. Did the author mean to say something like ‘Grafting of the plum and Myrobalan species is often characterized with the differentiation problem.’ (?)
3. Page 4, line 146: ‘…. fast growing ….’ => ‘…. fast growth …..’
4. Page 4, line 151-152: Sentence ambiguous, kindly rewrite to convey the intended message.
5. Page 6, line 224: ‘Ten%’ => ‘Ten percent’
6. Page 9, line 343: ‘recommended?’ => ‘recommended’
7. Page 12, line 510: ‘world widely’ => ‘worldwidely’
Author Response
We have complied with the minor comments as well as major comments. We rephrased the relevant parts, and more academically sound and precise terminologies are used, as mentioned. The title is complemented to reflect more to the Hungarian relations. Abstract has been expanded to better cover the actual content of the manuscript. The contents under the headings 4, 5, and 7 are revised, no longer contains seemingly redundant messages. Breeding, grafting compatibility and growing trends are addressed separately. Inclusion of authors’ perspective on potentially best rootstock for Hungary is included to the end of the conclusions.
Reviewer 3 Report
The manuscript deals with an interesting topic for the pomology sector and especially apricot tree cultivation. The manuscript presents significant information about the apricot rootstocks but there are many things that could be improved.
1. Although english is not my mother-language, I believe extensive english editing is required, as there are too many parts where the meaning is not so clear.
2. I believe the title should clearly indicate that the manuscript deals a lot with the conditions of apricot cultivation and rootstocks in Hungary along with the rest of the world, as there is too much information regarding apricot rootstocks use in Hungary.
3. there are some points in the manuscript which merit attention, as in Line 111 where it seems that VVA1 is not a Krymsj rootstock while the same is mentioned as Krymsk 1 in line 503, which is true.
4. There are some questionmarks found in the text, which should be deleted and the situation cleared out ((L188, 343)
5. Please state clearly when you speak about Mrobalan if you refer to seelings or clonal rootstocks (i.e. L171, 267 etc)
6. Be consistent with the name of the rootstocks (Torinel, Torifel etc, it is Torinel)
7. Please rephrase lines 260-264 in order for the meaning to be more clear (do you refer to fruits or leaves when you say the sugar content in the whole tree? - if you refer to leaves and not fruits, how is that related to the sugar content of the fruit?)
8. Based on my experience, apricot is not tolerant to lime conditions. In line 316 you state that apricot seedlings tolerate high lime content. Could you please cite more references for that. Furthermore, it would be great if the authors could limit hungarian bibliography to the least necessary, as most people do not know hungarian and in some cases there is only one reference supporting a statement and this is written in hungarian.
9. I recommend to state also the sucker producing habit of the rootstocks, as this is a valuable information
10, as this is a review manuscript, I urge the authors to build a Table where the properties of the rootstocks mentioned in their manuscript are presented.
Author Response
We have taken on board your comments and have rephrased the relevant parts. The English is checked not only by us, but by a professional too (native speaker). The title is complemented to reflect more to the Hungarian relations. The conflict of VVA 1 and Krymsk 1 is reviewed and resolved, such as the spelling of Torinel. Unnecessary question marks are removed, seedling or clonal propagation of Myrobalans is mentioned, when needed. The sugar content in question refers to fruit, and this has been made clearer in the text. In our experience, apricot does not tolerate lime, but it does tolerate slightly alkaline pH levels. This is what we understood and it is already fixed in the manuscript. In a short subsection, the suckering of rootstocks is also discussed.
Round 2
Reviewer 3 Report
The authors have responded to all my querries. I believe the manuscript can be accepted for publication by now.
Author Response
Dear Reviewer 3,
Thank You for Your time and efforts to review this manuscript. We took every suggestion, fixed every mistake, and improved the English. With Your helpful suggestions the text is more easyly readable and professional. We worked much on this project, and thank to Your expertise to suggest the final acceptance of this paper.
Yours sincerely,
the authors